# A Biosocial Perspective to Understand Antimicrobial Prescription Practices: A Retrospective Cross-Sectional Study from a Public Community Health Center in North India

**DOI:** 10.3390/antibiotics14030213

**Published:** 2025-02-20

**Authors:** Rashmi Surial, Sundeep Sahay, Vinay Modgil, Arunima Mukherjee, Ritika Kondal Bhandari

**Affiliations:** 1Society for Health Information Systems Programmes (HISP India), New Delhi 110025, India; rashmi.surial@hispindia.org (R.S.); vinay.modgil@hispindia.org (V.M.); arunimam@ifi.uio.no (A.M.); 2Department of Informatics and Centre of Sustainable Healthcare Education (SHE), Faculty of Medicine, University of Oslo, 0318 Oslo, Norway; 3Department of Pharmacology, Postgraduate Institute of Medical Education and Research, Chandigarh 160012, India; ritikakondal12@gmail.com

**Keywords:** antimicrobial resistance, prescription practices, drug quality, antimicrobial prescription, culture sensitivity testing, branded antimicrobials

## Abstract

**Background:** It is well established by research that large-scale and indiscriminate prescribing, dispensing, and use of antimicrobials drive antimicrobial resistance (AMR) endangering the health and well-being of people, animals, and the environment. In the context of low- and middle-income countries (LMICs), the prescribing of antimicrobials is often not based on biomedical rationality but involves alternative logic driven by social, cultural, and institutional factors. This paper seeks to develop a “biosocial” perspective, reflecting a unified perspective that treats the biomedical and social conditions as two sides of the same coin. **Methods:** This analysis is based on an empirical investigation of prescription slips that patients carry to buy drugs from the pharmacy following an outpatient department encounter with the clinician. Data collection involves mixed methods, including the quantitative analysis of the antimicrobials prescribed and a qualitative analysis of the underlying reasons for these prescriptions, as described by doctors, pharmacists, and patients. Data analysis involved triangulating quantitative and qualitative data, to develop a “biosocial” perspective, which can provide implications for the development of antimicrobial stewardship policies, particularly relevant for health institutions in low- and middle-income countries. **Results:** Our analysis of 1175 prescription slips showed that 98% contained antimicrobials, with 74% being broad-spectrum antimicrobials. Only 9% of cases were advised antimicrobial sensitivity testing (AST) before initiating treatment. Qualitative findings indicated that patients had poor awareness of antimicrobials and pharmacists played a crucial role in counseling. **Conclusions:** This study highlights that antimicrobial prescriptions in public health settings are influenced by both biomedical and social factors, supporting a biosocial perspective. Although AMS interventions are predominantly biomedical, adhering to clinical standards and best practices, this study underscores the necessity of integrating a biosocial viewpoint by incorporating the experiences of pharmacists and patient groups. Strengthening diagnostic support, patient education, and interprofessional collaboration could improve rational antimicrobial uses in low-resource settings.

## 1. Introduction

Inappropriate use of antimicrobials is magnified in the context of low- and middle-income countries (LMICs), where there are not only the formal means to access antimicrobials through doctor prescriptions but also multiple informal channels, such as over-the-counter sales, self-medication, and sharing of prescriptions and medicines with friends and families and through quacks [1,2,3]. This multiplicity brings diverse sociocultural implications in shaping processes of prescribing, dispensing, and consuming antimicrobials. These implications play out within institutional and regulatory frameworks and shape the efficacy of their implementation [4,5]. Understanding these multiple sociocultural–institutional implications in conjunction with biomedical information made visible by the prescription slip provides the basis to develop a unified “biosocial” perspective on the nature and determinants of antimicrobial prescriptions.

Our primary unit of analysis is the prescription slip, which has historically played a key role in defining medical practice and structuring the doctor–patient relationship. In LMIC settings, the prescription also structures the patient’s relationship with multiple others such as the pharmacist, who often substitutes for the doctor in providing advice to patients on what drugs to consume and how [6,7,8,9]. These relationships help shape access to therapeutics and allow for explorations of information and power asymmetries among doctors, patients, and pharmacists.

The research question that this paper pursues is “How does a biosocial approach help in understanding the nature of prescription practices and their determinants?” Empirically, the research is conducted within the context of an ongoing research study in India.

The rest of this paper is structured as follows. After this introduction, the next section briefly proposes the biosocial approach to help analyze the antimicrobial prescribing challenge. Following this, we provide details concerning the research methods (Section 7), and then present a case study description (Section 3 and Section 4). In Section 6, we present the analysis and discussion followed by brief conclusions.

## 2. The “Biosocial” Approach: For Unifying the Bio and Social Dimension of Antimicrobial Prescribing

### 2.1. The Biosocial Approach

Current streams of research on antimicrobial prescriptions are dominated by two relatively isolated streams of studies, one primarily biomedical and the other social. The biomedical stream focuses primarily on the determinants, mutations, and emergence of diseases [10,11], and in prescription studies, they seek to account for the pharmacological characteristics of antimicrobials [12]. An example of such an approach is presented in a recent paper by Shetty et al. (2024) who evaluated prescriptions across multiple tertiary facilities in India and analyzed deviations from standard treatment guidelines [13]. They evaluated 4838 prescriptions for their completeness (in terms of formulation, dose, duration, and frequency of administration), and how they adhered to standard treatment guidelines (STGs) developed by the Department of Health Research, ICMR, Government of India. Further, it was argued that these deviations had implications for drug interactions, lack of response, increase in cost of treatment, preventable adverse drug reaction (ADR), and antimicrobial resistance.

While such a biomedical analysis is crucial in understanding the therapeutic efficacy of drugs, the patient is missing from the analysis, who also plays a role in shaping not only what antimicrobials are prescribed, but also their dispensing and consumption. More recently, the social stream of research seeks to analyze the social, cultural, and political determinants of diseases to address this limitation. For example, from their studies in India, Charani et al. emphasized the importance of considering gender, caste, and literacy in such an analysis [14]. While this development represents an important step in incorporating the patient’s perspective, it does not adequately incorporate the biomedical aspects of the prescriptions.

Keeping these bio and social streams of research separate represents a dualism, as it is the same disease in focus highlighted through the prescription slip, which has combined influences of the biomedical and social. Focusing only on one creates one-sided causality arguments, which necessarily tend to be incomplete. To overcome this dualism and create a more duality-oriented perspective, Seeberg et al. developed the concept of the “biosocial”, which reflects an anthropological understanding that the bio and social are inseparable, evolving as complex intertwined ensembles over time [15]. The biosocial serves as an ontological position to analyze mutual and dynamic interactions between the biological and social contexts. Seeberg (2023) develops this perspective through the example of multidrug-resistant Tuberculosis (MDR-TB), which is showing alarming levels of increase, attributed not only to the biomedical condition but more closely to living conditions, population density, and environmental and sanitation conditions [15]. He argues for the development of a novel research paradigm that combines the bio and social, toward creating a more unified knowledge basis.

We develop this concept of the biosocial to analyze the interaction between biomedical and social elements, where the biomedical is understood through the drugs specified in the prescription slips and the social through discussions and interactions with clinicians, pharmacists, and patients. These two streams of analysis are then combined to understand the “what” and “why” of prescriptions to develop a more holistic biosocial approach. This represents a unique contribution of this research, which emphasizes that antimicrobials are not fixed medical objects but embody multiple constructed rationalities within different local contexts.

### 2.2. The Challenge of Antimicrobial Prescribing: Culture Matters

Antimicrobial prescribing presents a paradox for healthcare provision. Unnecessary use of antimicrobials can lead to resistance, harms patients, and increases treatment costs. Unjustified therapy with narrow-spectrum antimicrobials that ineffectively treat causative pathogens can also be detrimental to patients [16]. Excessive use of antimicrobials not only jeopardizes the clinical ability to treat and prevent microbial infections but also places fatal risks within many common medical procedures (such as C-sections) [17]. The prescription slip specifies the antimicrobials to use, thus playing a crucial role in managing the risks and opportunities of antimicrobials and their efficacy for care processes. Over, under, and non-optimal use of antimicrobials is shaped by social conditions, as argued by Charani (2022) as “culture matters” [14].

Unless we understand the sociocultural and behavioral drivers for antimicrobial use and develop contextually fit, equitable strategies to address them, no number of new antimicrobials will mitigate the rise in emerging antimicrobial resistance and its spread through populations [14,18].

Cultural influences emerge from multiple sources including medical specialists, national cultural characteristics, behavioral traits, and organizational policies and practices. Liu et al. analyzed the interconnections within a hospital setting between intrinsic and extrinsic factors, and, for example, noted female doctors were less likely to prescribe antimicrobials as they were more comfortable in adopting the “wait and see” principle, while also being more likely to succumb to patient requests for antimicrobials [19,20]. There are also national-level cultural differences, for example, Holland showed the lowest use of antimicrobials in Europe, while France, Belgium, Italy, and Germany were relatively higher [21]. Deschepper et al. drew upon Hofstede’s model of national cultural characteristics to analyze differences in antimicrobial prescribing across 14 European countries [22]. Countries with high uncertainty avoidance, which reflects a propensity to avoid uncertainty and risk, were more prone to prescribing antimicrobials, while Protestant societies, with values of austerity and simplicity, favored limited use of antimicrobials, and Catholic societies, more hierarchical and ritualistic, favored higher levels of antimicrobial use.

Behavioral aspects also shape antimicrobial prescribing practices. While junior doctors tended to rely more on broad-spectrum antimicrobials, senior doctors favored lower levels of prescribing. Charani et al. highlighted differences between medical specialists and clinicians, who may want to value their expertise and experience, as well as that of their peers [14]. In their study across 6000 households in Brazil, Marliere et al. found a widespread practice of patients storing unused drugs at home for use in future episodes of illness [23].

In India, overuse and non-use of antimicrobials are particularly influenced by the sale of non-prescription-based antimicrobials, despite legislation against it. The Government of India’s Policy for Containment of AMR (2011) to prevent over-the-counter sales of antimicrobials and making it mandatory for pharmacists to document their transactions is weakly implemented [24]. Similar has been the fate of the Government’s “Redline campaign” in 2016 to create public awareness of the rational use of antimicrobials to help curb self-medication. There is a current directive by the government for pharmacists to dispense antimicrobials in blue envelopes to distinguish them from other drugs. Patients, rather than obtaining a prescription from a doctor to access antimicrobials, adopt multiple other channels, to circumvent the prevailing regulations [25].

Various national-level guidelines have been developed by the National Center for Disease Control (NCDC) and the Indian Council of Medical Research (ICMR) to guide the use of antimicrobials [26,27] (Table 1). These guidelines have been difficult to implement, given the various resources and capacity constraints, and the different sociocultural influences in play.

We highlight the various biomedical and sociocultural conditions that influence antimicrobial prescriptions. We analyze them in unison through the biosocial perspective.

## 3. Results

### 3.1. Summary Statistics

A total of 1175 patients’ prescriptions were screened (Table 2), and their demographic features were summarized (Table 3).

### 3.2. Evaluation of Antibiotic Prescription Patterns

In all the prescriptions where an antibiotic is prescribed, in total, eight different antimicrobials were prescribed to various patients, with three accounting for more than 50% of the prescriptions. Amoxicillin + Clavulanic acid and Cefixime were the most prescribed (24% and 15%, respectively), followed by Azithromycin given to (11%) patients. Broad-spectrum antimicrobials accounted for 852 (74.02%) of the total prescriptions, as shown in Table 4 and Figure 1. In cases of multiple antimicrobial prescriptions, the second antimicrobials most given were Metronidazole and Cefixime. We also stratified the data based on gender. The top five antimicrobials given to female patients were Amoxicillin + Clavulanic acid, followed by Cefixime, Azithromycin, Ofloxacin + Ornidazole, and Ciprofloxacin. For male patients, it was Amoxicillin + Clavulanic acid, Cefixime, Azithromycin, Ciprofloxacin, and Doxycycline (Table 4).

### 3.3. Source of Prescriptions

The dental OPD was the biggest source of prescriptions (21%), followed by pediatrics (20%), general medicine (18%), dermatology (14%), and gynecology (8%), as shown in Figure 2.

### 3.4. Most Prescribed Antimicrobials for Symptomatic Treatment

We found the most common symptoms against which antimicrobials were prescribed were dental caries, respiratory conditions like fever, pneumonia, and other skin-related conditions like cellulitis, acne, UTI (urinary tract infections), and tonsillitis (Appendix A). The most prescribed antimicrobial for dental caries was Ofloxacin + Ornidazole followed by Amoxicillin and Amoxicillin + Metronidazole. For acute respiratory conditions, Amoxicillin + Clavulanic acid was the most prescribed along with Cefixime and Azithromycin. In cases of pneumonia, it was Amoxicillin + Clavulanic acid followed by Azithromycin, and for UTIs, it was Cefixime syrup for children and Nitrofurantoin for adults. For tonsillitis (Otolaryngological condition) it was Amoxicillin + Clavulanic acid followed by Azithromycin and Cefixime. For skin cellulitis, it was Amoxicillin + Clavulanic acid followed by Cefixime.

### 3.5. WHO AWaRe Classification

The WHO classified antimicrobials in AWaRe (Access, Watch, and Reserve) categories. Access includes the first- or second-choice antimicrobials that are most effective with the least potential for resistance. The Watch category includes the first- or second-choice antimicrobials used for a limited number of infections through empirical and non-targeted therapies [28]. We found five antimicrobials (Amoxicillin + Clavulanic acid, Nitrofurantoin, Doxycycline, Amoxicillin, and Metronidazole) prescribed among all the prescriptions from the Access category, which accounts for 62.76% of total prescriptions, and the remaining seven were Cefixime, Azithromycin, Ciprofloxacin, Ofloxacin-ornidazole, Cefpodoxime, Moxifloxacin, and Cefuroxime). Comprising 37% were from the Watch category. (No antimicrobials were prescribed from the Reserve category).

### 3.6. Incomplete Prescriptions

Antimicrobials sensitivity test (AST) reports were available in only 12 cases (1%) of slips studied and there were 106 cases (9.2%) where future tests were ordered. Some prescriptions were missing details of dosage (22.15%), duration of treatment (1.91%), and frequency of administration (2.86%). The diagnosis was often not mentioned, or the writing was illegible to understand. Further, the terminologies used for categorizing different signs and symptoms or diagnoses were highly variable. For example, fever with abdomen pain, burning micturition with abdominal pain, and tooth pain with extractions were all recorded as provisional diagnoses. More than 74% of prescriptions were related to broad-spectrum antimicrobials (e.g., Amoxicillin + Clavulanic acid, Ciprofloxacin, Doxycycline, and Cefixime), with more than 90% of patients being given antimicrobials empirically.

## 4. Qualitative Data Analysis

### 4.1. Theme 1: Poor Understanding Among Patients of What Are “Antimicrobials”

Most patients said that they had no idea about “what are antimicrobials”. Some mentioned that they did not care about the name of the medicine if it was freely available in the hospital and was effective in curing their illness. Another patient laughed and said, “It was the job of the doctor to know about medicines and not us”. From the 16 patients interviewed, only 1 male and 2–3 females had heard about antimicrobials. One female patient who was asked why she had been prescribed antimicrobials for long-term pain in her teeth, said that she was unaware about how antimicrobials could help. This response was similar to the reply of another female patient who had been prescribed antimicrobials for her child. Another male patient who was having a fever and was prescribed antimicrobials said he was prescribed some medicines for fever but had no idea that they were. There was another female patient, in her mid-twenties, who we observed buying antimicrobials from the pharmacist, and when asked what she was buying, “She was buying some medicines to treat an infection, and she knew about this since a doctor, a prior acquaintance of hers, had informed her that he was prescribing antimicrobials to be taken for 5 days. These observations were similar to a study conducted in Bahir Dar City, northwest Ethiopia, which found that only 48.5% of participants had moderate knowledge about antimicrobial resistance and usage [29]. In an another study, Hong et al. found that at an individual level, participants faced challenges in accessing healthcare knowledge and adhering to social expectations surrounding care [30].

We found two male patients, aged about 35 and 50 years who were diagnosed with an upper respiratory tract infection but had not been prescribed antimicrobials, much to their dismay and disappointment, since they believed then they would not be cured. Patients who had previously been prescribed antimicrobials for a particular health problem built expectations that for future health episodes of all kinds, they must be given antimicrobials. Similar to this, in a global survey by the World Health Organization (WHO), public knowledge and awareness about antibiotic use and antimicrobial resistance were found to be generally low, with many people holding misconceptions about the necessity and effectiveness of antimicrobials for various illnesses [31].

While most patients had little knowledge of what antimicrobials are or how they work, a few had some basic knowledge because of prior experience. Some patients believed that antimicrobials were a cure for all illnesses.

### 4.2. Theme 2: Limited Knowledge and Awareness of Antimicrobial Prescription Among Patients

When patients were asked whether they were counseled and told by the doctors about what was being prescribed to them, most said “no” and just told them to obtain the medicines written on the slip from the pharmacy. One patient coming out of the dermatology department said that the doctor had told him to buy the medicines and come back to him to understand how to take the medicines. We observed that the microbiologist and dermatologist shared a common room and talked to each other about patients, which may have resulted in the advice to patients to come back, which was not normal practice. Often, patients who were told to return after 7 days of treatment (or earlier if there was an emergency) never returned. A study in Nigeria found that many patients were not adequately counseled about their medications by doctors, similar to our observations, which potentially led to miscommunication and misuse of medications [32].

In our observations of patient–pharmacist interactions, we found it interesting to note the level of trust the patient had with the pharmacist, always soliciting opinions and advice on the drugs written in the prescription slip. One patient asked the pharmacist, “Can you give medicines like one for the gas (gastroenterology) problem you had previously given”. The pharmacist replied, “If you are not comfortable then you can skip these as they are not so important but do not tell the doctor that I gave you this advice”. Both shared a laugh, and the patients returned the medicines with a thank you.

Our observations clearly indicated that patients were reluctant to return to the doctor and clarify doubts about the drugs prescribed. When asked by a pharmacist why this was the case, the patient said, “If we went back to the doctor, he would get angry, so why don’t you please tell us”. The pharmacist said “This is their (the doctor’s) work on counseling patients and not ours”. But then he took their prescriptions and started telling them again about the drugs prescribed to them. Then, after the patient left the pharmacist told the researcher, “We do the work of the doctor also but still people will always complain about what work we have”. In one case, the patient was diagnosed with an upper respiratory tract infection and inquired with the pharmacist if they had been prescribed an antimicrobial or not. When the pharmacist replied “no”, the patient was very disappointed as he thought he would not be cured. As the patient was leaving, the researcher asked him how he knew about antimicrobials. He replied that he had a similar fever and cough in the past, and he was prescribed antimicrobials by the primary health care facility, which cured him in a few days. Such tendencies have also been reported from the UK, where patients often turned to pharmacists for advice on medication use, especially when reluctant to approach their doctors [33].

We observed an interesting difference between the parents of pediatric patients, who demanded prescriptions of branded antimicrobials from the doctors. Parents mentioned that “they can be careless in their own case and other adults in their family but preferred to buy branded antimicrobials for their children”. We saw in 71 slips from the pediatric OPD that branded antimicrobials were prescribed, such as Amoxicillin + Clavulanic acid and Cefixime syrup. Parents perceived the quality of medicines dispensed from the hospital pharmacy to be poor and so demanded branded medicines that they would procure from external pharmacies. The higher cost was not a consideration in such cases. This was also confirmed by the child specialist who prescribed branded medicines as they believed that the quality of medicines available in the hospital pharmacy was inferior to those available in the private pharmacies. To illustrate, the specialist asked the researcher to taste the vitamin C tablets available in the hospital dispensary and compare them with those available outside. He explained that “if the taste and quality of such a basic thing is poor inside the hospital then what can we expect from the other medicines”. The physician was hesitant to share further details about the drug quality and pointed toward the political pressures involved in the tendering of drugs in the hospital, and how this affected the quality of drugs purchased.

We developed short vignettes of nine patients with whom we followed up. In nearly all cases, we noted that when instructed by the doctor, the patient waited to obtain the AST report before commencing on the antimicrobials and tended to complete the dosage prescribed. In a few cases, patients discontinued antimicrobials midway through the course, either because they felt better or felt pain in the stomach when taking the drugs. One patient told us she was prescribed a 5-day antimicrobials course, but after 2 days did not feel better, and went instead to a private hospital for treatment. She attributed the problem to the doctor for not giving proper medicines. One patient, who had pus cells, but the AST showed a sterile sample, was still given antimicrobials as a precautionary measure. Another female patient who was pregnant was advised by her family members not to take excessive medicines since she was pregnant.

### 4.3. Theme 3: Pharmacists as a Repository of Administrative Information

We noted that the pharmacist, in nearly all cases, tells the patient about the drug, how to administer it, and what precautions are needed. For example, a pharmacist advised a patient prescribed Doxycycline to take it with at least a full glass of water and to clearly separate it from other medications such as those for blood pressure. While there was this sense of care seen with the pharmacists, they were often overworked with the customer load and complained that the doctors should do a better job in counseling the patients.

Most doctors expected that the pharmacist would explain to the patient the dosage, duration, and side-effects to patients. The pharmacist did this, but could not explain about the drugs themselves, which they thought should be explained by the doctors. The pharmacist, even if unaware about the drug, would give some basic information for building the satisfaction of patients, and direct them to the doctor for more information. However, the patients seldom approached the doctors because of fear. The important role of the pharmacist in counseling about antimicrobials has also been noted in other studies [34], and that pharmacists are often unable to provide comprehensive counseling due to inadequate knowledge and high workload [31].

The pharmacists commented on the positive development of how symptoms and diagnosis were now being mentioned on the prescription slips, which was not the case prior to the opening of the microbiology laboratory in 2021. The pharmacist showed a slip that had symptoms of cough and fever written, and a prescription of Azithromycin with Paracetamol for 3 days. We inferred that the pharmacist referred to signs and symptoms as the diagnosis. Common diagnoses (provisional/confirmed) noted on the slips included respiratory infections, tonsillitis, acute suppurative otitis media, urinary tract infections), cellulitis, abscess, and dental caries. We found 43% of the slips to be without diagnosis and based on signs and symptoms, such as pain in the tooth, sore throat, white discharge, pain in the abdomen, blurred vision, and wounds.

### 4.4. Theme 4: Doctors Justify Their Prescriptions on Medical Grounds

Doctors described the common ailments against which they prescribed antimicrobials to be upper respiratory tract infections, pneumonia, acute dysentery, tonsillitis, UTIs, skin issues (such as cellulitis, abscess, furuncle, and carbuncle), and dental cases (such as cellulitis, abscess and caries). It was interesting to see that most doctors said that they always ordered an AST before initiating antimicrobials, while our analysis indicated that was not the case. One doctor told us that he would often start with a first-line drug while waiting for the AST report and would change the regimen if resistance was detected. This often did not happen as patients did not frequently return for consultation. Another doctor said, “We can’t make the patient wait, we must give them something, and if we see pus in some patient then I order for an AST but also start with the empirical therapy”. Dental doctors did not order an AST and initiated antimicrobials as a precautionary measure.

The drugs that the doctors felt were most effective were Amoxicillin, Doxycycline, and Azithromycin, which differed from our analysis that showed the most prescribed drugs were Amoxicillin + Clavulanic acid, Cefixime, Azithromycin, and Ciprofloxacin, which belong to Amino penicillin + Beta-lactamase inhibitor, 3rd generation Cephalosporins, Macrolides, and Fluoroquinolones, respectively. Most doctors said that they prescribed drugs from the EDL, a point also confirmed by our analysis. While many physicians reported that they follow guidelines for AST, in practice there is a deviation due to time constraints and perceived patient expectations [35].

Doctors admitted that they had limited time to counsel patients and expected the pharmacists to do so. Noting that less than 20% of the patients who they advised to return for a follow-up visit actually came back, they implicitly suggested the futility of their counseling. Interestingly, we found differences in what was prescribed based on the prescriber, as shown in Table 5.

### 4.5. Theme 5: Limited Compliance of Prescriptions to National Guidelines

We found limited compliance of prescriptions with the biomedical guidelines provided by apex national medical research institutions of the Indian Council of Medical Research (ICMR) and the National Center for Disease Control (NCDC). This discrepancy is summarized in Table 6.

## 5. Phase 3 Analysis

### Building the Biosocial Perspective

While the quantitative data analysis reflected the biomedical parameters of what drugs were prescribed, the qualitative analysis clarified the perspectives of patients, clinicians, and pharmacists—the “why” underlying the prescriptions. In triangulating these two streams of analysis, we develop the “biosocial” perspective, reflecting that bio and social are not separate but two sides of the same coin. Given the significant diversity in sociocultural, economic, and health status across population groups, it is difficult to attribute any single reason for the diversity in prescriptions, but our approach helps to identify dominant tendencies on why certain antimicrobials were prescribed. Furthermore, information provided on prescription slips is often limited, such as missing information on confirmed diagnoses and prior history of prescriptions. However, observing the patterns over many slips and months provides an understanding of what antimicrobials are being prescribed and why (Table 7).

## 6. Discussion

### 6.1. Biosocial Themes

In public community health settings, prescription practices are severely influenced by biosocial components including both the social and biomedical dynamics. We discuss the different biosocial themes that emerged from our qualitative and quantitative analysis.

#### 6.1.1. Theme 1: Minimizing Biomedical Risks of Infections Through Broad-Spectrum Antimicrobials

Clinicians tend to mitigate risks in treating suspicious infections using broad-spectrum (>50%) antimicrobials such as Amoxiclav, Cefixime, and Azithromycin. This study supports the findings of Amritpal et al. (2018), who found that more than 65% of prescriptions were from broad-spectrum antimicrobials such as Amoxicillin–Clavulanic acid, Ceftriaxone, Ciprofloxacin, Clindamycin, and Piperacillin–Tazobactam [36]. The biomedical justification is that these drugs provide medical cover against a variety of pathogens, which enables clinicians to respond quickly to ambiguity surrounding bacterial infections, particularly created by the limitations of diagnosis and incomplete information of a patient’s medical history. This minimizes the risk of treatment failure, particularly in critical circumstances where delays in effective antimicrobial therapy may lead to serious complications or even death for patients.

While this approach helps minimize immediate clinical challenges, it brings up significant concerns regarding antimicrobial stewardship. From the demand side, a key social condition that promotes the significant use of broad-spectrum antimicrobials is the limited knowledge most patients have about antimicrobials and the implications [16,18]. Excessive and inappropriate use of broad-spectrum antimicrobials is a serious concern where diagnostic facilities are limited, which could lead to continued use of broad-spectrum agents. This inference emphasizes the need to both enhance diagnostic capacity and simultaneously build awareness of patients.

#### 6.1.2. Theme 2: Hospital Drugs Perceived to Be of Insufficient Quality for Treating Children

A critical issue concerns the perception of inadequate quality of antimicrobials available, particularly for pediatric patients in the hospital pharmacy. We found that 80 (11.8%) of the drugs prescribed from the pediatric OPD cases were branded medicines outside the EDL, such as Amoxicillin + Clavulanic acid and Cefixime syrup. One child specialist told us that there were political pressures involved in the tendering of drugs, to opt for the lowest-price generic drugs, which potentially compromised quality. A few parents told us that they may buy drugs from the hospital pharmacy for themselves and adult family members, which may be of suspect quality, but will not risk their children even given the attraction of lower-priced generic drugs. For children, they preferred the branded more expensive drugs from private pharmacies, which were abundantly available. Contributing to this poor perception may be the simple packaging and labels of generic drugs.

Even though 98% of prescriptions in the hospital aligned with the EDL in the hospital, many clinicians expressed concerns about the inadequacy of available formulations for effectively treating children. The lack of age-appropriate dosing or pediatric formulations may contribute to the reluctance of clinicians to prescribe these drugs. The high percentage of prescriptions based on generic names (91%) also raised concerns regarding the regularity and dependability of these formulations, even though they are inexpensive. The overall effectiveness of pediatric care may be impaired in situations when the quality of drugs is unknown, leading doctors to look for other treatments and potentially expose children to undesirable treatment plans. There is an urgent need to promote better access to high-quality pediatric drugs and their availability in the pharmacy. Addressing this crucial gap in healthcare delivery requires both the promotion of reliable pharmaceutical sources and improved training in pediatric pharmacology.

A similar study [37] found that nearly 25% of antimicrobials prescribed to children were inappropriate or unnecessary. This behavior can be seen as also being encouraged by the pharmaceutical nexus [38], where the medical representatives, in collusion with the doctors and pharmacists, promote certain branded medicines. Commercial interests complemented by weak regulation are often important drivers in the sale of sub-optimal-quality branded drugs. An important research need this raises is to study the quality of lower-priced branded drugs, to separate perception from reality.

#### 6.1.3. Theme 3: Invisibility of Infections Promotes the “Doctor Knows All” Attitude

A “doctor knows mindset” is dominant in a community hospital, as he/she tends to be most educated, and the patients are often unaware of the treatment protocols. The clinicians value highly their own clinical judgment and experience, and often believe that waiting for diagnostic results is a waste of time. While experience may be valuable in uncomplicated medical cases, when diseases become complex and uncertain, diagnosis-based evidence plays a crucial role in administrating less risky care protocols. In our analysis, we found only 9% of patients were advised to perform AST before antimicrobial prescriptions, indicating the prevailing belief among clinicians that their clinical judgment suffices for effective treatment. Another study reported that doctors’ attitudes and knowledge have a major effect on how they prescribe antimicrobials, and they highlighted the need to complement experience with clinical protocols [39,40]. Regarding conditions that are unusual or asymptomatic, relying on subjective assessment can be particularly detrimental and may result in a wrong diagnosis or incorrect administration of drugs.

This “doctor knows all” mindset is magnified by the invisibility of infections, particularly those that show no typical clinical signs, allowing physicians to avoid essential diagnostic tests. Another study identified several behavioral factors influencing antimicrobial prescribing, including perceived social pressure and the ease of making rational decisions about antimicrobial use [20]. Such an approach not only carries the risk of reducing the effectiveness of antimicrobial therapy, but also perpetuates trends in over prescribing broad-spectrum antimicrobials. What becomes important to understand is how clinical protocols and diagnostic support can complement doctors’ experience, particularly in the case of complex infections.

#### 6.1.4. Theme 4: Time Pressures of Doctors Limits Counseling of Patients

The high patient volumes experienced by clinicians create substantial time pressures that impede their ability to engage in meaningful patient counseling. This becomes a crucial handicap, given the poor knowledge and awareness of patients, of what are antimicrobials and how they should be consumed. Doctors have a serious lack of time for interactions with patients that leads to poor communication concerning treatment plans, potential negative outcomes, and neglecting the importance of following prescribed therapy. Despite their assertions that they offer counseling, many patients often leave appointments without an adequate understanding of their medications and why and how they should consume them. This information gap is often filled by pharmacists who provide advice to patients, particularly on administrative issues of how the drugs should be consumed but not on the biomedical characteristics of the drugs. This important piece of information thus does not come from either the doctor or the pharmacist, and patients are left largely on their own. This raises the importance of collaborative care approaches, involving doctors, pharmacists, and patients, for more effective treatment compliances [41,42].

This dependence of patients on pharmacists emphasizes the necessity of collaborative care approaches, in which pharmacists can be extremely important in medication management and patient education, which necessarily needs to be complemented with the doctor’s biomedical advice. The low levels of literacy and the high percentage (80% and more) of rural residents, made the patients often under-equipped to ask doctors these questions. Further, patients themselves often demanded antimicrobials from the doctor [43], raising the need for doctors to set aside adequate time to counsel patients. This is, however, easier said than implemented in practice.

#### 6.1.5. Theme 5: Follow-Up Visits by Patients Depend on Their State of Health and Social Advice, Often at the Cost of Defying Doctors’ Advice

We reported that only 30% of patients complete their antimicrobial course during follow-up visits, suggesting a worrying trend in which patient behavior is more affected by their own understanding of their health condition than by physician recommendations. Unless they have chronic problems or need continuous drugs, many patients stop taking drugs as soon as they start to feel better and frequently neglect returning for follow-up appointments. This reveals a serious absence of patient education leading to non-adherence of doctor’s advice.

Many patients told us that they were asked to do the AST before being prescribed antimicrobials. The treatment was often started without the AST, but after the culture reports, prescriptions were modified. We found that in a few cases, the patients stopped the medicines when they felt better, which often led to the recurrence of infections. Encouraging follow-up visits is crucial for monitoring patient progress and ensuring adherence to prescribed therapies [44,45]. In a few cases, on the phone call, the husband would reply on behalf of his wife, which could not let us ascertain first-hand the outcomes. This raises the possibility that patients fail to fully understand the justification for antimicrobial courses and the implications of ceasing medication abruptly. Better health outcomes and a reduction in the emergence of AMR could come from educational initiatives that emphasize the significance of follow-up visits, adherence to prescribed courses, and open communication from medical professionals.

#### 6.1.6. Theme 6: Symptoms Guiding Prescription of Antimicrobials

The clinical symptoms that drive antimicrobial prescriptions predominantly included fever, abdominal pain, burning micturition, and symptoms related to upper respiratory tract infections (URTIs). Clinicians often respond to these symptoms by prescribing antimicrobials, reflecting a pattern that prioritizes immediate symptom relief over evidence-based treatment protocols. Patients with prior histories of antimicrobial use are particularly influential in this dynamic, as they often demand antimicrobials based on their past experiences that enhances prescriptions [46], often inappropriate [47]. Our study raises similar concerns about the overuse of antimicrobials in certain OPD settings, especially dental, pediatrics, and skin. In many cases, such as dental, the antimicrobials are prescribed as a precautionary measure, or due to the issue of people not being able to afford definitive dental treatment, therefore acquiring recurrent infections.

We found next to no cases in which antimicrobial therapy was initiated after an AST. This reliance on symptomatic presentation exacerbates unwarranted prescriptions and is even seen in better-resourced settings such as Australia ([48]. We also found that in many cases of viral infections, antimicrobials were prescribed, as also noted in other studies for the treatment of URTIs [49]. During COVID-19, India saw large spike in the use of antimicrobials. To combat these trends, it is essential to foster a culture of diagnostic rigor that encourages healthcare providers to utilize appropriate testing and guidelines in making prescribing decisions. While this represents more of an idealized practice, it is difficult to implement given the resource and capacity constraints that exist. However, we believe our research can help flag these needs more strongly.

## 7. Material and Methods

### 7.1. The Research Context

The study was designed under a larger 4-year research project, one component of which was to understand how antimicrobial prescribing shapes AMR. The empirical focus was on a public health facility situated in a city in northern India with an estimated population of 13,200, which is two-thirds rural. The population there has a literacy rate of 93.31%, higher than the state average of 82.80%, with males at higher levels than women (95.63% to 90.87%).

This study was conducted at a 50-bed public community health center, serving 200–250 outpatients daily from the catchment population within a 4–5 km radius of the facility. The facility has a team of 13 doctors, covering ophthalmology, dentistry, medicine, dermatology, general medicine, microbiology, pediatrics, obstetrics, and gynecology. The hospital provides free biochemistry laboratory facilities, and in 2021 established a microbiology laboratory for antimicrobial susceptibility testing (AST) of urine, stool, and pus samples. There also is an inpatient ward, which is usually vacant, except in cases of emergency or during the rainy season. The facility hosts two pharmacies (dispensary and civil), with the dispensary within their premises, open from 9 am to 4 pm, providing free notified essential drugs. The civil pharmacy, located outside the premises, provides 24/7 services and a 10–15% discount on drugs. There are multiple private pharmacies just outside the premises, where patients can buy drugs perceived to be of higher quality but more expensive.

### 7.2. Research Design

This study was designed as a retrospective cross-sectional mixed-methods study. Ethics approvals and permissions from the senior medical officer (SMO) were taken before initiating the research. The pharmacist was also oriented towards the study. The data collection was conducted by a trained microbiologist, who was local to the area and participated in the development, pilot testing of the data collection tool, and trained in its deployment. Data collection involved multiple methods, which are described in the following sections.

### 7.3. Quantitative Study of Prescription Patterns

Prescription slips were collected from patients visiting the hospital pharmacy after an OPD encounter. The sample size was determined based on the availability of prescription slips within the study period (August 2022–July 2023). A total of 1175 prescription slips were randomly selected from patients visiting the pharmacy after an OPD encounter using a systematic daily random sampling approach, capturing antimicrobial prescription across different OPDs. The researcher stood outside the pharmacy and, based on a random sampling method, approached patients (irrespective of their age and gender) and after taking their verbal consent, selected their slips that included antimicrobials. About 2–3% of the patients coming to the pharmacy were approached each day during the period August 2022–July 2023, to gain insights into seasonal variations. The inclusion criteria consisted of prescription slips from outpatients receiving at least one antimicrobial, ensuring the study remained focused on antimicrobial prescribing practices. Prescriptions without antimicrobials, illegible or incomplete prescriptions lacking essential details, and inpatient prescriptions were excluded to maintain data relevance and quality. A sample of prescription slips is provided in Appendix A.

The images of selected prescriptions were captured, while anonymizing the patient’s name, by the researcher using her mobile phone. The image was then studied to extract relevant details such as the patient’s age, sex, location of residence; the prescription date, name of antimicrobials, whether branded or generic, and if from the essential drugs list; symptoms or confirmed diagnosis; whether an AST was ordered; legibility of the doctor’s handwriting; and any other follow up notes and advice (see Appendix A). These details were recorded in an Excel sheet and then imported into a database to develop the analytics.

### 7.4. Qualitative Study to Understand the “Why” of Prescriptions

Observations and discussions were carried out with patients, pharmacists, and clinicians to understand the “why” of the prescriptions, from their individual perspectives. During informal interactions with patients, we sought to understand their level of awareness about what antimicrobials were, why they believed they were prescribed, and how they should be consumed. The researcher also conducted observations of patient–pharmacist interactions, to understand the kind of queries the patient had and the responses received from the pharmacist. Issues were further discussed with 16 patients (6 males and 10 females all in the age group 25–50 years), 7 doctors (3 general practitioners, 1 each from dental, microbiology, dermatology, and anesthesia), and 2 pharmacists. Further, telephonic follow-ups after due consent were conducted with 9 patients who had visited the microbiology laboratory for a culture test and had been prescribed antimicrobials. They were asked if they had complied or not with the prescription regime, whether they had collected their test reports, and what they did with them. These different discussions and observations were recorded by the researcher in her field diary.

### 7.5. Study Objectives and Outcomes

The main objective of the study was to assess antibiotic prescribing practices in conjunction with biosocial determinants. Important outcomes include the percentage and categorization of prescribed antimicrobials, adherence to the WHO AWaRe classification and national guidelines (ICMR, NCDC), the prevalence of empirical versus culture-based prescriptions, and the incidence of AST. The study also analyzed social, cultural, and institutional factors affecting prescribing practices through qualitative interviews and observations. This holistic approach seeks to enhance antimicrobial stewardship efforts in resource-constrained settings.

### 7.6. Data Analysis

The data analysis process, oriented toward developing a biosocial perspective on prescription practices, was conducted in three phases. The first was the phase of quantitative data analysis, to discern the biomedical patterns underlying prescribing. The data recorded in the Excel data sheet were imported into a database system, to first develop summary statistics to characterize the data, and then analyzed concerning with respect to the WHO core parameters (such as the AWaRe classification: average number of antimicrobials dispensed per encountered/visit, % of drugs prescribed by generic name, % of encounters with an antimicrobial prescribed), national guidelines, and other indicators such as compliance with the essential drug lists (EDLs). The second was the qualitative data analysis that first involved compiling all the textual data in a Word file, and then the next involved the reading of this text file by the different authors to discern their respective themes, which were then discussed to agree on the final themes. As a third step, the quantitative and qualitative analyses were combined to interpretively understand how the identified qualified themes could be related to the biomedical patterns discerned through the quantitative analysis. This triangulation helped to develop the biosocial perspective on prescription patterns.

## 8. Conclusions

Our study is among the first of its kind in India that presents a biosocial perspective on prescription patterns in a community health context and offers important insights into the link between integrated biological data and the social determinants influencing antimicrobial prescriptions. This study highlights that antimicrobial prescribing in public health settings is shaped by both biomedical and social factors, with a high reliance on broad-spectrum antimicrobials (74%), low adherence to national guidelines, and limited use of antimicrobial sensitivity testing (9%), leading to empirical prescribing. While quantitative analysis uncovers trends and frequently prescribes antimicrobials, qualitative data describes the social dynamics that impact prescription decisions by elucidating the attitudes and reasons of patients, physicians, and pharmacists. The results highlight how the perceptions of patients and clinicians about infections and treatment, their time constraints, and the use of broad-spectrum antimicrobials impact prescribing practices that might not be in accordance with optimum policies for antimicrobial stewardship. To improve antimicrobial stewardship, there is an urgent need to enhance diagnostic support, strengthen prescription oversight, integrate pharmacists into stewardship programs, and increase awareness among clinicians and patients. Although AMS interventions are predominantly biomedical, adhering to clinical standards and best practices, our study underscores the necessity of integrating a biosocial viewpoint by incorporating the experiences of pharmacists and patient groups. Despite limitations in resources and time, the long-term advantages—such as enhanced adherence, patient engagement, and sustainable stewardship—may surpass the associated costs. Implementing these measures can optimize antimicrobial use, improve prescription practices, and mitigate the growing threat of AMR in resource-limited settings.

## Figures and Tables

**Figure 1 antibiotics-14-00213-f001:**
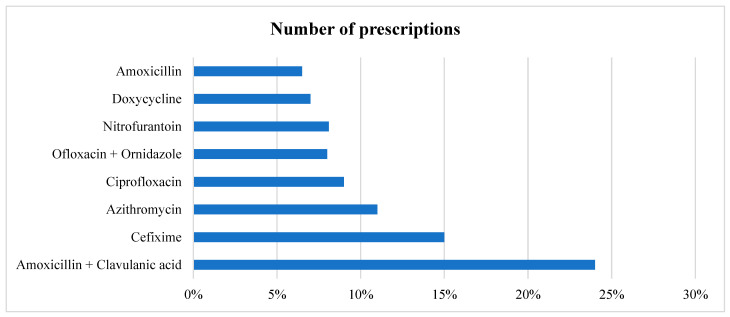
Prescription pattern of different antimicrobials.

**Figure 2 antibiotics-14-00213-f002:**
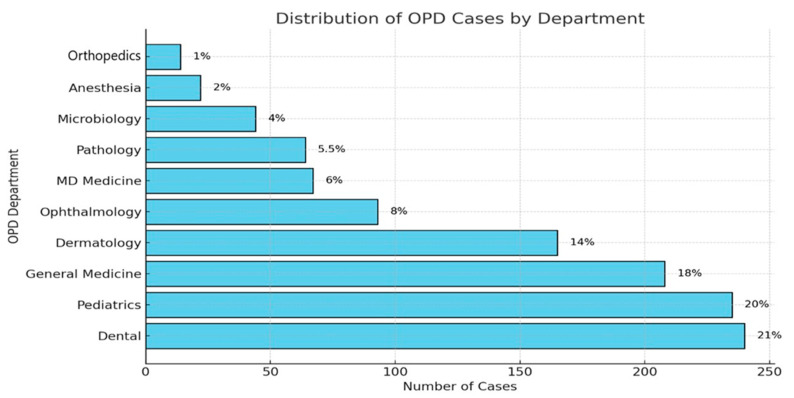
OPD-wise breakdown of antimicrobial prescriptions.

**Table 1 antibiotics-14-00213-t001:** Key features of national antimicrobials prescribing guidelines.

S. No.	ICMR Guidelines	NCDC Guidelines
1.	Make a clinical diagnosis before starting treatment.	Send the patient to follow up on standard investigations to make the correct diagnosis
2.	Limiting empirical treatment of antimicrobial therapy.	Antimicrobials should be started only after sending culture, if facilities are available.
3.	Knowing your bug before starting antimicrobial treatment.	Assessing the factors affecting the activity of antimicrobials.
4.	Choosing appropriate antimicrobials for treatment and modifying them depending on culture-sensitive reports	Review antimicrobial therapy and it should be escalated or de-escalated accordingly after receipt of culture report.

**Table 2 antibiotics-14-00213-t002:** Summary statistics of prescription slips screened.

Parameter	N (%)
Total prescription slips	1175
Total prescriptions with antimicrobials	1151 (98%)
Total prescriptions with provisional diagnosis	24 (2%)
Total antimicrobial prescriptions included in EDL	1134 (98.5%)
Prescriptions based on generic names	1050 (91%)
Prescriptions with multiple antimicrobials	135 (12%)
Average range of number of drugs in a slip	1 to 3

**Table 3 antibiotics-14-00213-t003:** Demographic characterization of patients.

Demographic Parameter	Characteristic N (%)
Gender	Male 525 (43%) Female 650 (57%)
Age group (in years)	1–5 years: 180 (15.6% 6–15 years: 164 (14%) 16–39 years: 386 (34%) 40–65 years: 325 (24%) Above 65: 96 (8.4%)
Patient location	Rural: 771 (67%) Urban: 380 (33%)
Most common comorbidities	Acute respiratory infections, caries, and urinary tract infections
Most prescribed antimicrobials	Amoxicillin + Clavulanic acid, Cefixime, Azithromycin
Most prescribed antibiotic classes	Aminopenicillins + Clavulanic acid, 3rd generation Cephalosporins, Macrolides
Broad-spectrum antimicrobials	Amoxicillin + Clavulanic acid, Cefixime, Azithromycin, Ciprofloxacin, Ofloxacin–Ornidazole, Doxycycline
Narrow-spectrum antimicrobials	Nitrofurantoin and Amoxicillin
WHO AWaRe compliance (common drugs)	**Access:** Amoxicillin + Clavulanic acid, Nitrofurantoin, Doxycycline, Amoxicillin; **Watch:** Cefixime, Azithromycin, Ciprofloxacin, Ofloxacin–Ornidazole; **Reserve:** No drugs

**Table 4 antibiotics-14-00213-t004:** Distribution of prescribed antimicrobials by class, spectrum, and AWaRe category.

Name of Antimicrobials	Number of Prescriptions	Class of Antimicrobials	Spectrum	AWaRe Category
Amoxicillin + Clavulanic acid	275 (24%)	Aminopenicillins + Clavulanic acid	Broad	Access
Cefixime	170 (15%)	3rd generation Cephalosporins	Broad	Watch category
Azithromycin	131 (11%)	Macrolides	Broad	Access category
Ciprofloxacin	100 (9%)	2nd generation Fluoroquinolones	Broad	Watch category
Ofloxacin + Ornidazole	90 (8%)	Aminopenicillins	Broad	Watch category
Nitrofurantoin	89 (8.1%)	Nitrofurans	Narrow	Access category
Doxycycline	86 (7%)	Tetracycline	Broad	Access category
Amoxicillin	75 (6.5%)	Aminopenicillins	Narrow	Access category

**Table 5 antibiotics-14-00213-t005:** Differences are based on who prescribes.

S. No.	Conditions Mentioned by Doctors Requiring Antimicrobial Prescriptions	Antimicrobials Prescribed by Medical Doctors (with MBBS)	Antimicrobials that Doctors (with Higher MD Degrees) Think Need to Be Prescribed	Antimicrobials that Were Mostly Prescribed as Seen in the Data
1.	RTI	Azithromycin or Amoxicillin or Doxycycline	Amoxicillin	Amoxicillin + Clavulanic acid
2.	UTI	Nitrofurantoin	Nitrofurantoin	Syrup Cefixime (for children) Nitrofurantoin
3.	Tonsillitis	Azithromycin	---	Amoxicillin + Clavulanic acid
4.	Skin Cellulitis	Doxycycline or Amoxicillin + Clavulanic acid	Amoxicillin + Clavulanic acid	Amoxicillin + Clavulanic acid
5.	Pneumonia	Amoxicillin or Doxycycline or Azithromycin	Amoxicillin + Clavulanic acid or Cefixime	Amoxicillin + Clavulanic acid
6.	Fever	Azithromycin or Doxycycline	Doxycycline	Azithromycin

**Table 6 antibiotics-14-00213-t006:** Compliance of prescriptions with national guidelines.

ICMR Guideline	NCDC Guidelines	Levels of Compliance Based on Data
Make a clinical diagnosis before starting any treatment.	Send the patient for follow-up on standard investigation for correct diagnosis	The clinical diagnosis (presumptive or confirmatory) was made in 52% of the slips and the other 48% were given antimicrobial treatment based on signs and symptoms.
Limiting empirical treatment of antimicrobial therapy.	Antimicrobials should be started only after sending the appropriate culture if facilities are available.	From the 681 slips analyzed, 93 were sent for urine and 25 for pus culture. All patients advised for pus culture were started with empirical treatment of Amoxicillin + Clavulanic acid. Out of 93 patient slips advised with urine culture, 77 were prescribed empirical treatment with Nitrofurantoin (for adults) or Cefixime (for children). Only in 16 cases, the doctor waits for the AST report.
Knowing your bug before starting antimicrobial treatment.	Assessing the factors affecting the activity of antimicrobials.	ASTs are rarely conducted before empirical therapy
Choosing appropriate antimicrobials for the treatment and modifying treatment based on AST results.	A review of antimicrobial therapy must be done and escalated or de-escalated based on the culture report.	There were only 11 such cases (out of 93 patient slips for urine culture) where the AST report led to the escalation or de-escalation of the therapy.

**Table 7 antibiotics-14-00213-t007:** Comparative analysis of biomedical and social perspectives on prescriptions *.

Main Themes	Biomedical Perspective	Social Perspective of Clinicians	Social Perspective of Patients	Social Perspective of Pharmacists	Interpretation of the Biosocial
Antimicrobial prescription	Most prescribed drugs: Amoxiclav, Cefixime, Azithromycin	What the doctors say: Amoxicillin, Doxycycline, Azithromycin	Limited idea about the content of the prescription	Most dispensed drug Amoxiclav for almost all ailments	Minimizing risks through initiating treatment with broad-spectrum antimicrobials
Order for AST culturing tests	Only 9% were advised	They say they order AST before prescribing	Limited idea about the need for AST	ASTs are mostly performed only in the case of illness recurrence	ASTs not advised because of the invisibility of infection and that the “doctor knows”
Compliance with the essential drug list	98% of the prescriptions were from EDL	Doctors have adequate guidelines	No information on what the EDL is	The list is well followed in this facility as there are sufficient medicines in the health setting	The EDL includes 14 antimicrobials, which is sufficient for most cases, except for children
Generic drugs	91% of prescriptions are based on generic names	Doctors prescribe branded drugs only when not available in hospital pharmacy	Their priority is to only obtain free drugs	Only stock-branded drugs	Prescriptions based primarily on access to drugs
Counseling of patients	Limited counseling performed by doctors for lack of time	They say they counsel	Receive drug administration information from pharmacists	The doctors should do it, but do not, forcing them to provide information	Social interactions between doctors (pediatrics and dermatology) promoted a limited degree of counseling
Follow-up treatment	30% of the patients completed their course after follow-up	Patients complete the course and report back	Course taken until they felt better and very few reported back to the doctor	Only chronic patients report back as they need to obtain their monthly stock of medicines	Doctor’s advice on follow-up is generally ignored
Symptoms guiding prescription of antimicrobials	The most common symptoms are fever, abdominal pain, burning micturition, tooth extraction	URTI, UTI, abscess, or when there is a visible infection	Patients with a prior history of using antimicrobials demand more, most have no idea what antimicrobials are	Tonsillitis, respiratory conditions, bronchitis, not viral	Antimicrobials are prescribed to manage changing patterns of infections

* This table provides the basis for developing relevant biosocial themes.

## Data Availability

The datasets used and/or analyzed during the current study are available from the corresponding author upon reasonable request.

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
