# Peer review of "A Biosocial Perspective to Understand Antimicrobial Prescription Practices: A Retrospective Cross-Sectional Study from a Public Community Health Center in North India"

_antibiotics, 2025, doi:10.3390/antibiotics14030213_

Round 1

Reviewer 1 Report

Comments and Suggestions for Authors

A biosocial perspective to understand prescription practices: Analysis from a public community health centre in North India

Reviewer Report

This study examines antimicrobial prescribing practices in low- and middle-income countries from a biosocial perspective. It highlights that prescribing practices that contribute to antimicrobial resistance are shaped not only by biomedical factors, but also by social and institutional influences. The study integrates quantitative and qualitative methods by analyzing outpatient prescription records taken to pharmacies. The findings provide valuable insights for the development of antimicrobial stewardship policies in low- and middle-income countries. Although the study provides a solid overall framework, there are scientific and formal aspects that need improvement. My detailed opinions are as follows:

1)     Prospective, retrospective, questionnaire etc. study type should be added to the title.

2)     The results and conclusions in the abstract are not fully clear and should be included. A conclusion that responeses the study’s aim should be provided.

3)     The references need to be revised in accordance with the journal’s guidelines, using square brackets.

4)     Sections 1 and 2 comprehensively and adequately present the significance and rationale of the study. A reference to the guideline mentioned in Table 1, line 154, should be provided.

5)     Was a sample size calculation performed for the study? How was the sample determined? In addition, on what basis were the inclusion and exclusion criteria established?

6)     What was used in the statistical analyses?

7)     Tables and figures should be revised according to the journal’s guidelines. Abbreviations in the table should be stated below the table.

8)     Tables should be listed in the Results section. Please move Tables 6, 7, and 8 to the Results section.

9)     Although the discussion section is organized thematically, the integration of the findings with clinical relevance and their connection to the literature is inadequate. Please revise the Discussion to clearly and specifically address and respond to the purpose of the study. Once these revisions are made, the discussion section will be more advanced.

10) More concrete conclusions/recommendations should be provided to directly address the study aim.

11) References should be revised according to the journal’s guidelines. It has been observed that some references lack details such as volume, issue, page numbers, and titles. These should be carefully checked and revised accordingly.

Comments on the Quality of English Language

Minor English grammar errors should be corrected.

Author Response

S. No.

Questions

Answers

1.

Prospective, retrospective, questionnaire etc. study type should be added to the title.

Thank you for your valuable suggestion regarding specifying the study type in the title. After examining the study design, we have concluded that our study is best categorized as a retrospective cross-sectional mixed-methods study. So based on this we have modifying the title to:

“A Biosocial Perspective to Understand Prescription Practices: A Retrospective Cross-Sectional Study from a public community health centre in North India”. Please refer to line number 2 to 4 and is highlighted in the revised manuscript.

2.

The results and conclusions in the abstract are not fully clear and should be included. A conclusion that responeses the study’s aim should be provided.

As per suggestions, we have incorporated the results and conclusions in the abstract part in the revised manuscript. Please refer to line number 27-39 and is highlighted.

3.

The references need to be revised in accordance with the journal’s guidelines, using square brackets.

As per journal’s guidelines, references has been modified in the revised manuscript.

4.

Sections 1 and 2 comprehensively and adequately present the significance and rationale of the study. A reference to the guideline mentioned in Table 1, line 154, should be provided.

As per suggestions, reference has been provided to the guidelines mentioned in the Table 1 in the revised manuscript. Please refer to line number 164-165 and is highlighted.

5.

Was a sample size calculation performed for the study? How was the sample determined? In addition, on what basis were the inclusion and exclusion criteria established?

In this study, the sample size was determined based on the availability of prescription slips within the study period (August 2022 – July 2023). A total of 1,175 prescription slips were randomly selected from patients visiting the pharmacy after an OPD encounter using a systematic daily random sampling approach, capturing antimicrobial prescription across different OPDs. The inclusion criteria consisted of prescription slips from outpatients receiving at least one antimicrobial, ensuring the study remained focused on antimicrobial prescribing practices. Prescriptions without antimicrobials, illegible or incomplete prescriptions lacking essential details, and inpatient prescriptions were excluded to maintain data relevance and quality.

As per suggestions, changes have been made in the methodology section regarding sampling criteria. Please refer to line number 202-216 and is highlighted in the revised manuscript.

6.

What was used in the statistical analyses?

This study primarily used descriptive analysis of 1,175 prescription slips. Data were recorded in Microsoft Excel, and frequencies and percentages were calculated to summarize antimicrobial prescription patterns, drug distribution, and compliance with WHO AWaRe classification. Comparative assessments were made across age groups, gender, and clinical departments using simple frequency distributions. For qualitative data, thematic analysis was conducted to identify key insights from interviews and observations. As the study was focused on descriptive analysis of prescribing patterns, no statistical testing was performed.

7.

Tables and figures should be revised according to the journal’s guidelines. Abbreviations in the table should be stated below the table.

As per suggestions, we have revised all tables and figures to align with the journal’s formatting guidelines.

8.

Tables should be listed in the Results section. Please move Tables 6, 7, and 8 to the Results section.

As per suggestion, Table 6,7, and 8 has been moved to result section in the revised manuscript.

9.

Although the discussion section is organized thematically, the integration of the findings with clinical relevance and their connection to the literature is inadequate. Please revise the Discussion to clearly and specifically address and respond to the purpose of the study. Once these revisions are made, the discussion section will be more advanced.

Thank you for your insightful feedback. To improve clarity and integration, we have moved the qualitative findings from the discussion section to the results section to ensure a more structured and concise presentation. This adjustment shortens the discussion in the revised manuscript.

Additionally, we have incorporated more recent citations to strengthen the connection between our findings and existing literature. In the revised discussion, we have combined both qualitative and quantitative data to develop a biosocial theme, effectively linking social and biomedical dynamics influencing antimicrobial prescription patterns. Please refer to line number 507-651.

10.

More concrete conclusions/recommendations should be provided to directly address the study aim.

As per suggestions, the conclusion part has been modified in the revised manuscript. Please refer to line number 662-683 and is highlighted.

11.

References should be revised according to the journal’s guidelines. It has been observed that some references lack details such as volume, issue, page numbers, and titles. These should be carefully checked and revised accordingly.

As per suggestions, we have revised the references to ensure full compliance with the journal’s guidelines. Missing details, including volume, issue, page numbers, and titles, has been carefully checked and updated for accuracy and consistency in the revised manuscript.

Reviewer 2 Report

Comments and Suggestions for Authors

Line 181-182: Images are missing

Line 185-186: The ethical approval may be provided as supplementary data with certification that ethical approval is as per Indian legislations.

Line 242: Table 2, The range of antibiotics numbers may be written instead of 2.5.

Line 243: Table 3, please check the values of age group 6-15 years.

It is recommended that mathematical calculations of this table and others may be checked again.

The discussion part is not up to the mark. The results and background of the study has been included in this section and need to be included in their respective sections for better readability of the manuscript. Moreover, authors have not discussed their results or findings in the light of already published data or probable reasoning of the same. The authors have written criticizing language to the target group under study and may be omitted.

It is also suggested to add the prescriptions written after hiding confidential details as supplementary data.

Author Response

#Reviewer 2:

Q. No.

Questions

Answers

1.

Line 181-182: Images are missing

Correction has been made in the revised manuscript. These images were removed from the manuscript earlier from editor’s comments.

2.

The ethical approval may be provided as supplementary data with certification that ethical approval is as per Indian legislations.

Thank you for the suggestion. As per suggestions, we will provide the ethical approval certificate as supplementary data.

3.

The range of antibiotics numbers may be written instead of 2.5.

As per suggestions, range of average number of prescribed antibiotics has been provided in the table 2. Please refer to line number 272 and is highlighted in the revised manuscript.

4.

Table 3, please check the values of age group 6-15 years.

As per suggestions, correction has been made in the table. Please refer to line number 275 (Table 3) and is highlighted in the revised manuscript.

5.

Line 243:

It is recommended that mathematical calculations of this table and others may be checked again.

Thank you for your suggestion. We have rechecked all mathematical calculations in the tables to ensure accuracy throughout the manuscript.

6.

The discussion part is not up to the mark. The results and background of the study has been included in this section and need to be included in their respective sections for better readability of the manuscript. Moreover, authors have not discussed their results or findings in the light of already published data or probable reasoning of the same. The authors have written criticizing language to the target group under study and may be omitted.

Thank you for your insightful feedback. To improve clarity and integration, we have moved the qualitative findings from the discussion section to the results section to ensure a more structured and concise presentation. This adjustment shortens the discussion in the revised manuscript. Additionally, we have incorporated more recent citations to strengthen the connection between our findings and existing literature. In the revised discussion, we have combined both qualitative and quantitative data to develop a biosocial theme, effectively linking social and biomedical dynamics influencing antimicrobial prescription patterns. Please refer to line number 507-651.

This study does not aim to find faults but rather provides an evidence-based assessment to support policy improvements and antimicrobial stewardship. The discussion is written neutrally and objectively, focusing on systemic factors affecting prescribing patterns rather than blaming any specific group.

7.

It is also suggested to add the prescriptions written after hiding confidential details as supplementary data.

Thank you for your suggestion. As per suggestions, we have provided the anonymized prescription slips as supplementary data.

Reviewer 3 Report

Comments and Suggestions for Authors

Abstract:

The abstract effectively summarizes the article in a clear and concise manner.

Introduction:

all references in the Bibliography section are in the text, but the following references  in the introduction are in the text but not in the Bibliography in the fallowing lines:

79-80 - Charani et al, 2021.

88 and 90 - Seeberg et al 2020

124 - Liu, et al 2019b

line: 130, 141, 143: references are from 2000, 2001 and 2008, too old to be used as a comparison, check for updated references.

line 121, text: (p. 1506) (8), appropriate reference missing in Bibliography.

Discussion 

reduce the discussion section, some conversations between the authors and the people in the study are not relevant to the article. 

missing text reference to Table 5.

Figure 1 and Table 5 should be in the supplement section.

Author Response

#Reviewer 3:

Q. No.

Question

Answer

1.

Introduction:

all references in the Bibliography section are in the text, but the following references in the introduction are in the text but not in the Bibliography in the fallowing lines:

79-80 - Charani et al, 2021.

88 and 90 - Seeberg et al 2020

124 - Liu, et al 2019b

line: 130, 141, 143: references are from 2000, 2001 and 2008, too old to be used as a comparison, check for updated references.

line 121, text: (p. 1506) (8), appropriate reference missing in Bibliography.

Thank you for your feedback. We have added missing references (Charani et al., 2021; Seeberg et al., 2020; Liu et al., 2019b) to the Bibliography, updated old references (2000, 2001, 2008) with recent ones, and corrected the citation format for Line 121. All references have been added according to journal’s guidelines.

2.

reduce the discussion section, some conversations between the authors and the people in the study are not relevant to the article. 

Thank you for your insightful feedback. To improve clarity and integration, we have moved the qualitative findings from the discussion section to the results section to ensure a more structured and concise presentation. This adjustment shortens the discussion in the revised manuscript. Additionally, we have incorporated more recent citations to strengthen the connection between our findings and existing literature. In the revised discussion, we have combined both qualitative and quantitative data to develop a biosocial theme, effectively linking social and biomedical dynamics influencing antimicrobial prescription patterns. Please refer to line number 507-651.

However, we believe that the conversations between the researcher and the people in the study are important as they provide contextual insights into prescribing behaviors. These qualitative findings help in understanding the biosocial dynamics influencing antimicrobial use, which is crucial for shaping effective policy recommendations and antimicrobial stewardship interventions.

3.

missing text reference to Table 5.

Text has been provided in the text for table 5 in the revised manuscript to Table S1 (Supplementary data). Please refer to line number line no. 302 and is highlighted.

4.

Figure 1 and Table 5 should be in the supplement section.

As per suggestions, figure 1 and table 5 has been moved to supplementary data in the revised manuscript.

Reviewer 4 Report

Comments and Suggestions for Authors

1- you need to indicate the study type in the method section, is this a retrospective study, cross-sectional or what?

2- is it not clear what are the outcomes of the study? that is an essential component and should be included in the method section to provide the reader with what are you studying 

3- Table 1 should include more details on the baseline characteristics, including comorbidities, antibiotics prescribed, etc. How can we assess the population of the study without including Table 1

4- the design and objectives of the study are not clear and confusing. I cannot read the discussion since these details are not clear!

Author Response

#Reviewer 4:

Q. No.

Questions

Answers

1.

you need to indicate the study type in the method section, is this a retrospective study, cross-sectional or what?

This study was designed as a retrospective cross-sectional mixed-methods study. This text has been provided in the method section of revised manuscript. Please refer to line number 195 and is highlighted in the revised manuscript.

2.

is it not clear what are the outcomes of the study? that is an essential component and should be included in the method section to provide the reader with what are you studying 

Thank you for your insightful comment. To ensure clarity, we have stated the study outcomes in the Methods section to align with the study objectives. Please refer to line number 243-249 and is highlighted.

3.

Table 1 should include more details on the baseline characteristics, including comorbidities, antibiotics prescribed, etc. How can we assess the population of the study without including Table 1

As per suggestions, Table 1 has been modified with suggested details.

4.

the design and objectives of the study are not clear and confusing. I cannot read the discussion since these details are not clear!

Thank You for your suggestions, we have modified the study design and objectives to ensure better readability. The Methods section now explicitly defines the study as a retrospective cross-sectional mixed-methods study, detailing both the quantitative and qualitative components, sampling criteria etc. Additionally, we have clearly outlined the study objectives, specifying the key outcomes related to antimicrobial prescription patterns, adherence to guidelines, and biosocial factors influencing prescribing behaviors. We have also shortened the discussion part in the revised manuscript. These revisions ensure a clear and structured foundation for the interpret the findings in the context of the aim of the study.

Round 2

Reviewer 1 Report

Comments and Suggestions for Authors

Thank to the authors for their revisions. The study has improved significantly. I just have one suggestion: the abstract has become too long. Please provide a more concise and focused conclusion. I have no additional comments.

Author Response

Thank to the authors for their revisions. The study has improved significantly. I just have one suggestion: the abstract has become too long. Please provide a more concise and focused conclusion. I have no additional comments.

Ans; As suggested, we have revised the abstract to make the conclusion more concise and focused, ensuring that it clearly reflects the key findings without unnecessary detail. Thank you for your valuable insights throughout the revision process. Please refer to line number 34-40 in the revised manuscript.

Reviewer 2 Report

Comments and Suggestions for Authors

It is suggested to add the prescriptions as supplementary data after hiding confidential details of patients only.

In India, the primary authorized body for ethical permission regarding medical data collection is the Indian Council of Medical Research (ICMR). As per its "National Ethical Guidelines for Biomedical and Health Research involving Human Participants" the approval from an Institutional Ethics Committee (IEC) within the jurisdiction of research institution is required to collect medical data adhering to the ICMR guidelines.

The authors have provided approval letter from HISP India Research Ethics Advisory Committee on the letter pad of HISP INDIA (Society for Health Information Systems Programmes). The HISP INDIA (Society for Health Information Systems Programmes) appears not to be a registered and authorized body for granting ethical permission. A letter from ICMR certifying that HISP INDIA (Society for Health Information Systems Programmes) is authorized body to grant ethical permissions for medical data collection may be provided as supplementary data.

Author Response

  1. It is suggested to add the prescriptions as supplementary data after hiding confidential details of patients only.

Ans; As per suggestions, we have provided prescription slips with hiding confidential details in the revised manuscript.

  1. In India, the primary authorized body for ethical permission regarding medical data collection is the Indian Council of Medical Research (ICMR). As per its "National Ethical Guidelines for Biomedical and Health Research involving Human Participants" the approval from an Institutional Ethics Committee (IEC) within the jurisdiction of research institution is required to collect medical data adhering to the ICMR guidelines. The authors have provided approval letter from HISP India Research Ethics Advisory Committee on the letter pad of HISP INDIA (Society for Health Information Systems Programmes). The HISP INDIA (Society for Health Information Systems Programmes) appears not to be a registered and authorized body for granting ethical permission. A letter from ICMR certifying that HISP INDIA (Society for Health Information Systems Programmes) is authorized body to grant ethical permissions for medical data collection may be provided as supplementary data.

ANS; This project is part of a four-year study funded by the Research Council of Norway, with HISP India serving as a research partner responsible for coordinating the empirical work. Memorandums of Understanding (MoUs) have been established with the participating hospitals where the empirical work is being conducted. Dr. Gitika Arora, the lead at HISP India, has overseen the ethical agreements for this study. The approval letter from HISP India Research Ethics Advisory Committee has been attached. However, as the participating public health facility does not have its own Institutional Ethics Committee (IEC), we obtained the necessary permissions from the Senior Medical Officer (SMO) of the hospital, along with the hospital administration, ensuring full access to facilities and resources to conduct the research. Additionally, PGIMER Chandigarh, which has an established IEC, is a collaborating institution in this study. Ethical approval for the broader project has been obtained from PGIMER's IEC. Furthermore, the project underwent formal review and approval by the Health Ministry Screening Committee (HMSC), Government of India, with approval granted during the HMSC meeting held on January 30, 2023 (A approval mail has been given below).

Reviewer 4 Report

Comments and Suggestions for Authors

None

Author Response

R4;

No comments

We sincerely thank Reviewer 4 for their time and effort in reviewing our manuscript. Your thoughtful evaluation and consideration are greatly appreciated. Your review process has been invaluable in refining our work, and we truly appreciate your time in assessing our study. Thank you for your support.